# Mutational signature dynamics indicate SARS-CoV-2's evolutionary capacity is driven by host antiviral molecules

**Kieran D. Lamb** [1,2☯], **Martha M. Luka** [1,2☯], **Megan Saathoff** [1], **Richard J. Orton** [1], **My V. T. Phan** [3,4], **Matthew Cotten** [1,3,4,5], **Ke Yuan** [2,6,7]*, **David L. Robertson** [1]*

**1** Medical Research Council - University of Glasgow Centre for Virus Research, School of Infection and Immunity, Glasgow, Scotland, United Kingdom, **2** School of Computing Science, University of Glasgow, Glasgow, Scotland, United Kingdom, **3** Medical Research Council/Uganda Virus Research Institute and London School of Hygiene & Tropical Medicine Uganda Research Unit, Entebbe, Uganda, **4** College of Health Solutions, Arizona State University, Phoenix, Arizona, United States of America, **5** Complex Adaptive Systems Initiative, Arizona State University, Scottsdale, Arizona, United States of America, **6** School of Cancer Sciences, University of Glasgow, Glasgow, Scotland, United Kingdom, **7** Cancer Research UK Scotland Institute, Glasgow, Scotland, United Kingdom

☯ These authors contributed equally to this work.
* Ke.Yuan@glasgow.ac.uk (KY); David.L.Robertson@glasgow.ac.uk (DLR)

**Data Availability Statement:** Computational code is available at https://github.com/kieran12lamb/SARS-CoV2_Mutational_Signatures GISAID data accessions are available at doi.org/10.55876/gis8.

## Abstract

The COVID-19 pandemic has been characterised by sequential variant-specific waves shaped by viral, individual human and population factors. SARS-CoV-2 variants are defined by their unique combinations of mutations and there has been a clear adaptation to more efficient human infection since the emergence of this new human coronavirus in late 2019. Here, we use machine learning models to identify shared signatures, i.e., common underlying mutational processes and link these to the subset of mutations that define the variants of concern (VOCs). First, we examined the global SARS-CoV-2 genomes and associated metadata to determine how viral properties and public health measures have influenced the magnitude of waves, as measured by the number of infection cases, in different geographic locations using regression models. This analysis showed that, as expected, both public health measures and virus properties were associated with the waves of regional SARS-CoV-2 reported infection numbers and this impact varies geographically. We attribute this to intrinsic differences such as vaccine coverage, testing and sequencing capacity and the effectiveness of government stringency. To assess underlying evolutionary change, we used non-negative matrix factorisation and observed three distinct mutational signatures, unique in their substitution patterns and exposures from the SARS-CoV-2 genomes. Signatures 1, 2 and 3 were biased to C→T, T→C/A→G and G→T point mutations. We hypothesise assignments of these mutational signatures to the host antiviral molecules APOBEC, ADAR and ROS respectively. We observe a shift amidst the pandemic in relative mutational signature activity from predominantly Signature 1 changes to an increasingly high proportion of changes consistent with Signature 2. This could represent changes in how the virus and the host immune response interact and indicates how SARS-CoV-2 may continue to generate variation in the future. Linkage of the detected mutational signatures to the VOC-defining amino acids substitutions indicates the majority of SARS-CoV-2's evolutionary capacity is

221201qs, doi.org/10.55876/gis8.230406qg and doi.org/10.55876/gis8.230406fb.

**Funding:** The authors acknowledge funding from the Medical Research Council (MRC, MC_UU_12014/12 to DLR, MC_UU_00034/5 to DLR and a Doctoral Training Programme in Precision Medicine studentship for KDL, MR/N013166/1 to KY and DLR), the Wellcome Trust (220977/Z/20/Z to MC, KY, DLR), the UK Department for International Development (DFID) under the MRC/DFID Concordat agreement (MC_PC_20010 to MC), Engineering and Physical Sciences Research Council (EPSRC, EP/R018634/1 to KY), and the European Union's Horizon 2020 research and innovation programme project PANCAIM (101016851 to KY). The funders had no role in study design, data collection and analysis, decision to publish, or preparation of the manuscript.

**Competing interests:** The authors have declared that no competing interests exist.

likely to be associated with the action of host antiviral molecules rather than virus replication errors.

## Author summary

We show that both public health measures and virus properties are associated with the rise and fall of regional SARS-CoV-2 reported infection numbers with regional differences attributable to the extent of vaccine usage and the effectiveness of public health measures. In our mutational signature analysis, using non-negative matrix factorisation, we detected three distinct mutational signatures that can be putatively attributed to the action of specific host antiviral molecules. Interestingly, we observe a shift in mutational signature activity from predominantly Signature 1 changes to an increasingly high proportion of changes consistent with Signature 2. These mutation patterns influence SARS-CoV-2's evolutionary capacity, the available genetic variation that selection can act on, and so can be linked to the mutations defining the variants of concern responsible for the distinct SARS-CoV-2 infection waves. The dominant types of nucleotide substitutions involved indicate that much of the mutation and hence variation come from the action of the host immune response rather than replication errors since the virus has an error correction system.

## Introduction

The COVID-19 pandemic began in late 2019 following a zoonotic spillover event of a SARS-related coronavirus, subsequently named SARS-CoV-2, in Wuhan, China [1, 2]. The extensive and rapid global spread of this new human coronavirus and its detrimental impact on human health has rendered it among the most significant pandemics in recent history [3]. Different geographical regions of the world have reported varied infection patterns that are attributed to differences in population demographics and health care systems, diverse government responses [4, 5], the emergence of more transmissible variants [6, 7] and other viral, human and population factors. Since its emergence, SARS-CoV-2 has undergone significant genetic change such that numerous variants, i.e., distinct genotypes, have been identified [8], many with altered phenotypic properties [9].

The World Health Organization (WHO) and other public health bodies have broadly classified variants that pose an increased risk to global public health (due to increased transmissibility, increased virulence or decrease in the effectiveness of public health measures relative to 2019/early 2020 SARS-CoV-2 variants) as variants of concern (VOCs) and variants of interest (VOIs) [10]. The early SARS-CoV-2 variants to emerge in 2019 and the more transmissible +S:D614G variant followed by the VOCs (Alpha, Beta, Gamma, Delta and currently Omicron) have driven significant and sequential "waves" of SARS-CoV-2 infections internationally. The emergence of each variant showing a clear geographical link [11–13].

Viral mutations arise from a diverse set of processes (principally viral polymerase replication errors and host anti-viral editing processes), which can be identified by the characteristic mutational signatures that they leave on the genome [14, 15]. Such characterisation of dominant mutational processes is routinely used in cancer genomics [16]. The catalogue of SARS-CoV-2 nucleotide changes show distinct mutational patterns suggestive of a role for host antiviral mutational processes in introducing changes in the viral RNA [17, 18]. These

processes potentially dominate in SARS-CoV-2 evolution because point mutations introduced in replication are mostly corrected by the action of a proofreading enzyme.

The generation of virus diversity, the key to virus persistence by generating novel variation and thus evolutionary capacity, is multi-faceted [19], yet our understanding of the relative importance of underlying mutational processes linked to the action of host anti-viral molecules is still very limited. Given that SARS-CoV-2 continues to develop new variants, many associated with sets of previously observed (convergent) and novel mutations [9], it is critical that we improve our understanding of the mechanisms and sources of evolutionary change.

Along with routine surveillance of SARS-CoV-2 infections, there has been an unprecedented global sequencing effort resulting in databases containing many millions of genome sequences, in particular GISAID [20]. Here we examined this data to describe the global molecular epidemiology and evolution of SARS-CoV-2. Using regression models we first examined how viral properties and public health measures have influenced the magnitude of infection waves in different geographic locations. Satisfied that SARS-CoV-2 variants have been an important driver of infections we then used non-negative matrix factorisation to characterise the mutational processes involved in the generation of variants and their changing patterns of activity over time.

## Results

### Characterising the SARS-CoV-2 waves regionally

This first part of the study reports on global SARS-CoV-2 data from 24/12/2019 to 28/01/2022 only as limited public health measures were in place after this time. We observed 1,544 distinct SARS-CoV-2 lineages from 7,348,178 sequences. 88% of the infections in the global pandemic during this time frame were caused by a subset of 13 Pango and WHO variants (S1 Table). While there are geographical differences there is a clear dominance of a subset of variants and replacement of these through time (Fig 1). This "wave" infection pattern was evident in all geographic locations. Although biased by testing rates, Europe and the Americas had the highest infection rates, reporting up to 450 cases per million population per day (Fig 1). The emergence or introduction of VOCs coincided with a steep increase in infection rates globally. For example, cases in Asia showed a steep rise in February 2021, which peaked in May 2021 (Fig 1, **panel Asia**). During this period, Alpha and Delta comprised greater than 75% of the SARS-CoV-2 cases identified in the sequence data. Africa and Oceania on the other hand displayed overall sustained low case numbers. Despite this, Beta dominated the second wave in parts of Africa while Alpha dominated the third Oceanic wave. After its emergence in March 2021, Delta spread to become the predominant variant across all continents. The Omicron variant of concern was first identified in South Africa in late November 2021 and, by January 2022, it had rapidly become the predominant cause of infections worldwide (Fig 1).

### Covariates of the waves

We investigated the degree to which public health measures and viral properties explain continent-specific reported cases of infection. Correlation analysis at the global level showed a significant correlation between infection rates and the predictor variables: government stringency, vaccination, previous infection burden, virus diversity and fitness (S2 Table).

Regression analysis revealed that the impact of the predictor variables on the magnitude of reported cases were found across all continents. We classified significance levels as follows: no significance for p-values greater than 0.05, weak significance for p-values between 0.05 and 0.001, and high significance for p-values less than 0.001. Our findings indicated that

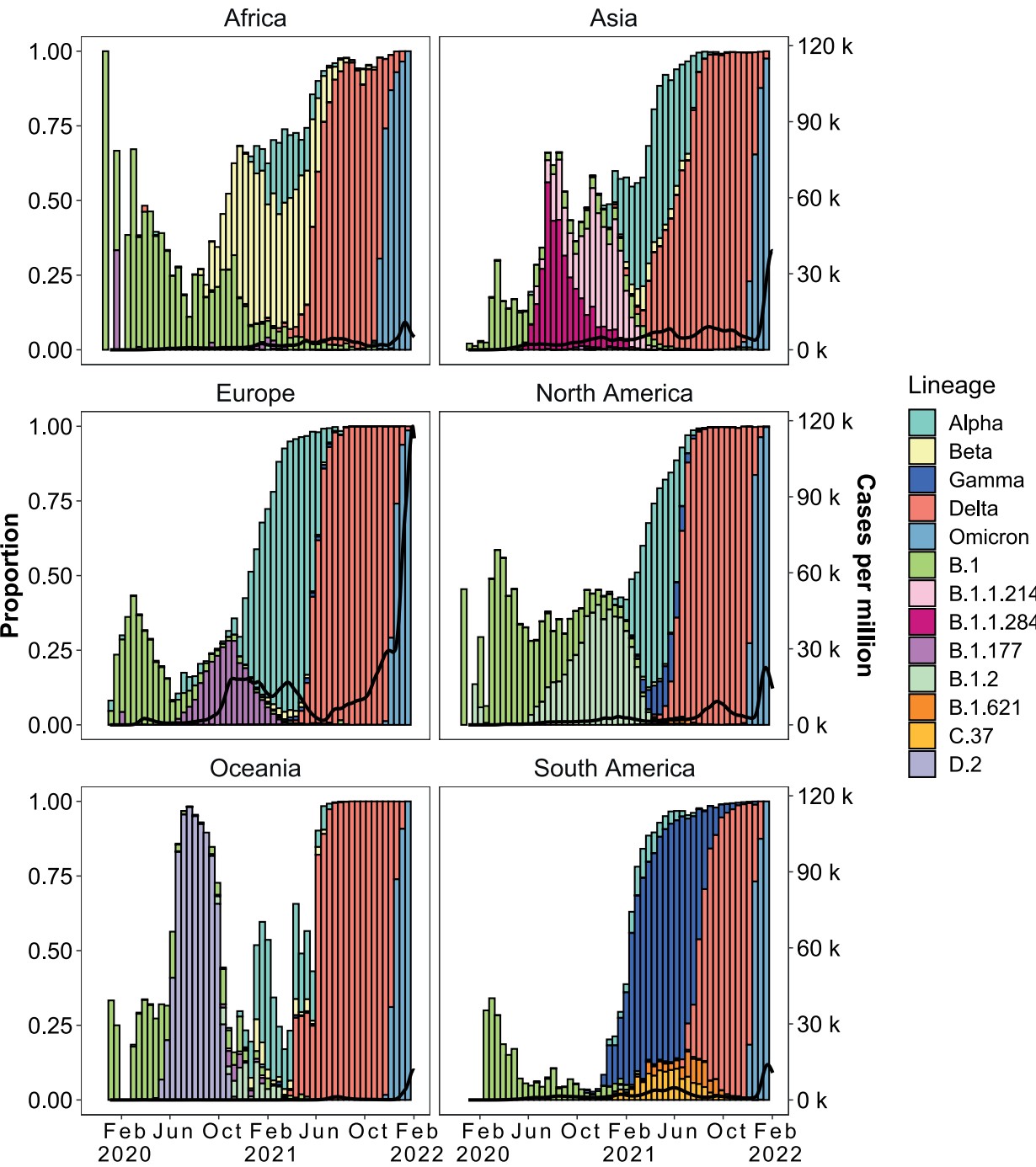

**Fig 1. Continent-level SARS-CoV-2 lineage dynamics and pandemic curves.** Lines show a 14-day rolling average of reported SARS-CoV-2 cases. Bars show the biweekly proportions of common lineages and are coloured by lineage. The white space shows the proportion of sequences from other (non-majority) lineages.

government stringency had a weakly significant impact in Asia, Europe, and South America, but a strongly significant impact in Africa, Oceania, and North America. Virus fitness, previous infection burden, and vaccination demonstrated a strongly significant impact across all continents. Virus diversity was strongly correlated with high infection numbers in Europe and

North America, with a weaker association in Africa, Asia, Oceania, and South America. The R squared values, indicating the proportion of variance explained by our model, were greater than 0.5 for all continents, ranging from 0.66 in Oceania to 0.79 in Africa (S3 Table). Generally, our predictions closely resembled the rise and fall of SARS-CoV-2 infection case numbers (Fig 2).

For country-level analysis, we included 29 countries from six continents based on the completeness of data (availability of sequence data in every 14 day bin). Pandemic plots were visualised using biweekly bins and multiple linear regression was fitted using the same approach. Different countries had varying lineage dynamics as illustrated in S1 Fig. The five predictor variables had varying impacts on infection rates across countries (S2 Fig). Despite some differences related to the population level processes investigated here, there is a clear variant replacement process taking place. As the generation of novel variants is fundamentally a mutation dependent process we next investigated the underlying patterns of mutations being generated through time. The goodness of fit varied among countries, with the R squared varying from 0.28 (Japan) to 0.96 (Australia), with a median of 0.69 (S4 Table). Though our model successfully captured the general infection wave patterns in many countries, it struggled to capture short-term data spikes in specific instances, such as in Belgium (November 2020), India (May 2021), Indonesia (August 2021) and Japan (September 2021) (S2 Fig).

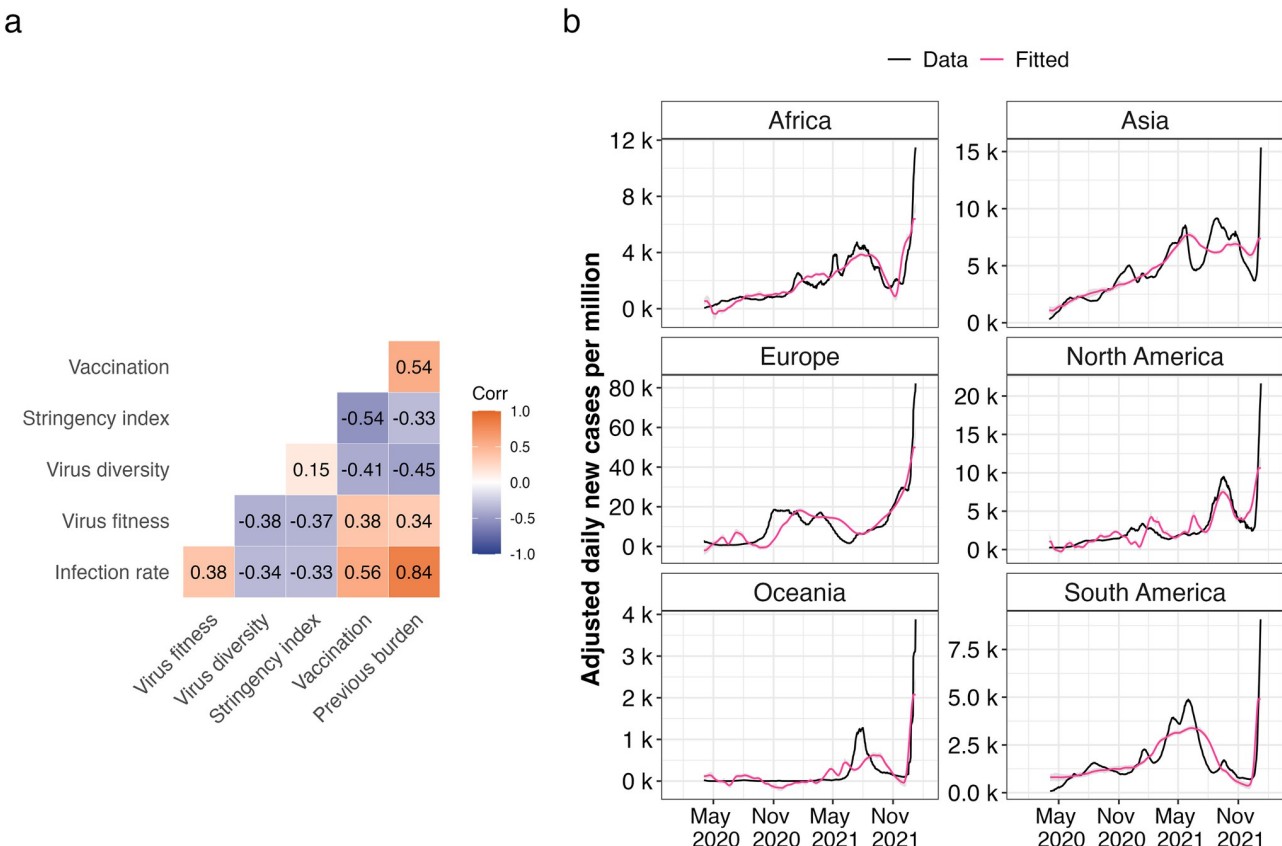

**Fig 2. Association of SARS-CoV-2 infection rates and predictor variables globally. A**. Pearson's correlation matrix of infection rate and predictor variables. Positive correlations are denoted in orange and negative correlations in blue and colour intensity is directly proportional to coefficient value. **B**. Model fitting using multiple linear regression. Black solid lines show a 14-day rolling average of adjusted SARS-CoV-2 cases. Pink solid lines show fitted mean response values of infection rates with predictor values as input.

## Identifying putative mutational processes contributing to changes in SARS-CoV-2

New variants of concern have displaced viral lineages that were previously dominant in the population in different geographical regions and in some cases globally (Fig 1). This behaviour has been observed with the original variants of concern (Alpha, Beta and Gamma) and then globally with the Delta and Omicron lineages. We investigated whether these variant wave events (periods of time where infections are dominated by a single variant) were linked to the activity of specific mutational processes. Each of the variants of interest/concern has evolved independently such that detecting the patterns of mutations in the SARS-CoV-2 sequence data allows us to observe which processes are most active and could be contributing to the emergence of variants.

Mutations were called using inferred references for each of the Pango lineages, which we call tree-based referencing (S3 Fig). The SARS-CoV-2 alignment of 13,278,844 sequences up to 26/10/2022 was used. Of these 13 million sequences 2,195,182 sequences were selected as they contained 5,726,144 newly arisen mutations. Cytosine to thymine mutations (C→T) were the most common and were the primary substitution category for most weeks where sequences were recorded. Note, SARS-CoV-2 has an RNA genome but we refer to uracil as a thymine to match pre-existing DNA mutational signature notations.

Three signatures were identified with distinct substitution patterns using non-negative matrix factorisation (NMF) (Fig 3 and S5 Fig). Signature 1 is heavily biased towards C→T mutations. Signature 1 had a high probability of ACA, ACT and TCT contexts (adjacent nucleotides in the 5' and 3' direction of the mutated site), consistent with what was earlier reported by Simmonds et al. [17] as highly mutated contexts for C→T substitutions in SARS-CoV-2. Signature 2 is predominantly adenine to guanine (A→G), guanine to adenine (G→A) and thymine to cytosine (T→C) mutations. The proportion of A→G and T→C mutations is approximately equal in this signature, which is indicative of a double-stranded mutational process. SARS-CoV-2 mutations at adenine positions on the negative strand will be counted as thymine mutations due to the negative strand being used to replicate positive sense RNA, with the mutated A→G now pairing with a cytosine on the +sense RNA and replacing the original thymine [21, 22]. Signature 3 is predominantly composed of guanine to thymine (G→T) substitutions.

## The dynamics of mutational processes through the pandemic

By using the available SARS-CoV-2 sequences we can measure the mutational signature activity across time as long as our samples are aggregated using time series annotations. Signature exposures (Fig 4) show that Signature 1 remained the most prominent signature throughout the pandemic, although following the emergence of Signature 2 its activity reduced proportionally. Absolute exposure values (Fig 4B) show that Signature 1 does not appear to reduce its exposure, rather Signature 2 increases its exposure. Signature 2 establishes itself as a substantial signature after December 2020. It continues to expand after October 2021, just prior to the emergence of the Delta VOC. Signature 3 is by far the least active of the three signatures but remains consistent until after January-February 2022 when it begins to drop towards zero. This is around the time Omicron began to emerge as the dominant VOC.

Combined signature activity reached a peak between July and October 2021 (Fig 4B) coinciding with the peak number of unique mutations (Fig 5A and 5B). This is around the time the mutational signature dynamics appear to be shifting, with Signature 2 contributing more unique mutations. We can see that this also coincides with the Delta VOC wave, which, between May 2021 and January 2022, was the lineage group showing the greatest number of

## Signature 1 (Normalised)

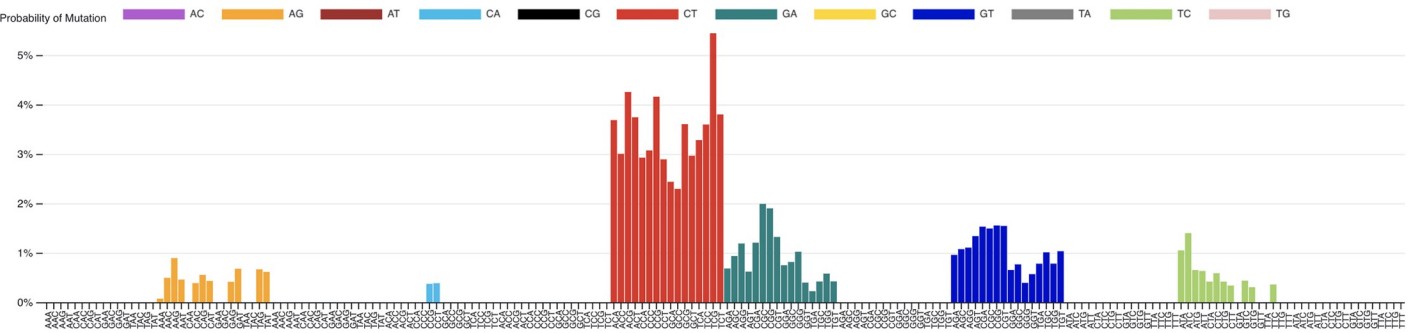

## Signature 2 (Normalised)

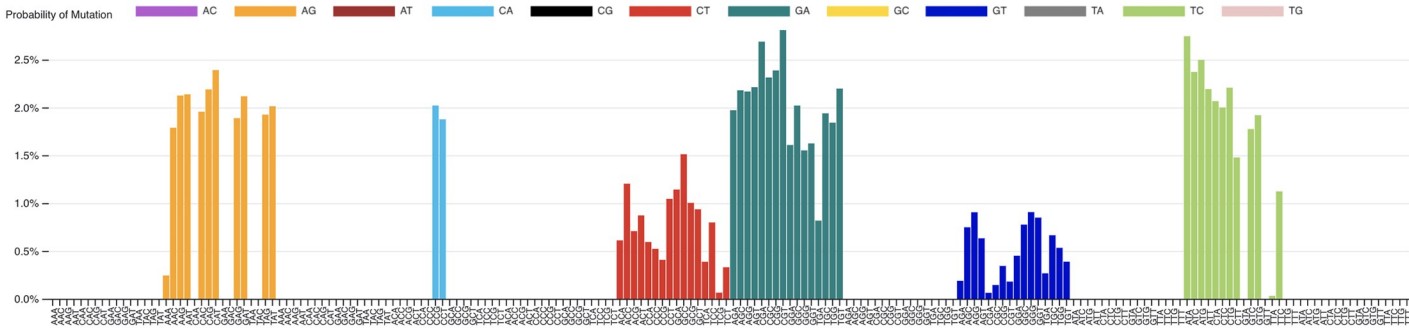

## Signature 3 (Normalised)

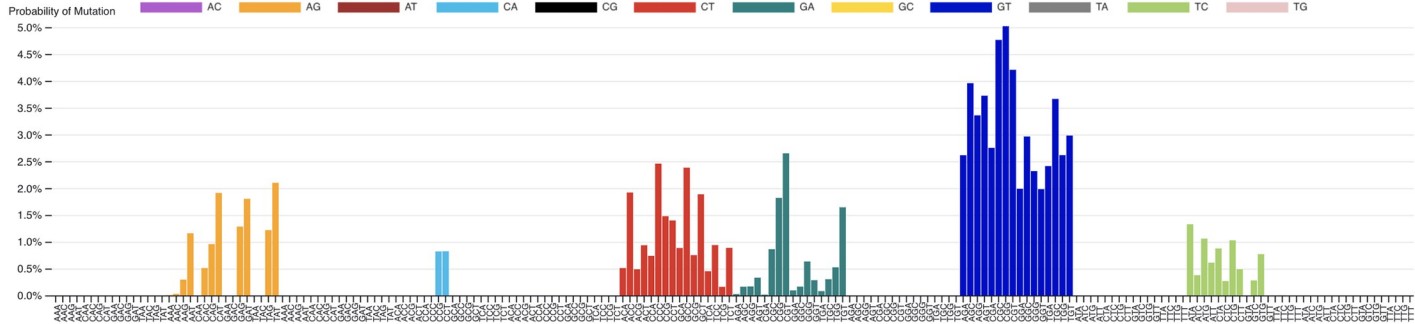

**Fig 3. Mutational signatures extracted from the SARS-CoV-2 genome sequences by non-negative matrix factorisation.** Signatures are patterns of probabilities for each category of substitution in a three nucleotide context. Each bar represents a context and is coloured by the substitution category of the mutation that occurs there. Each signature may represent a distinct mutational process. Signature 1 is heavily biased towards cytosine to thymine (C→T) mutations, particularly in 3' CpG contexts TCG, CCG and ACG. Signature 2 from SARS-CoV-2 is predominantly adenine to guanine (A→G), guanine to adenine (G→A) and thymine to cytosine mutations (T→C). Signature 3 is strongly guanine to thymine (G→T), a pattern that is thought to be caused by the action of guanine oxidation by reactive oxygen species. Signatures are shown normalised against the tri-nucleotide composition of the SARS-CoV-2 genome. Non-normalised forms in the context of the SARS-CoV-2 genome composition are shown in S5 Fig.

newly acquired mutations (Fig 5). Delta was the first VOC to dominate on a global scale, outcompeting other VOCs like Alpha, Beta and Gamma in their regions of circulation. Omicron similarly repeated this phenomenon, almost entirely replacing Delta globally within weeks of its emergence (Fig 5B). We also see a marked decrease in the activity of Signature 3 following Omicron's establishment as the dominant variant. A similar decrease in G→T mutations was

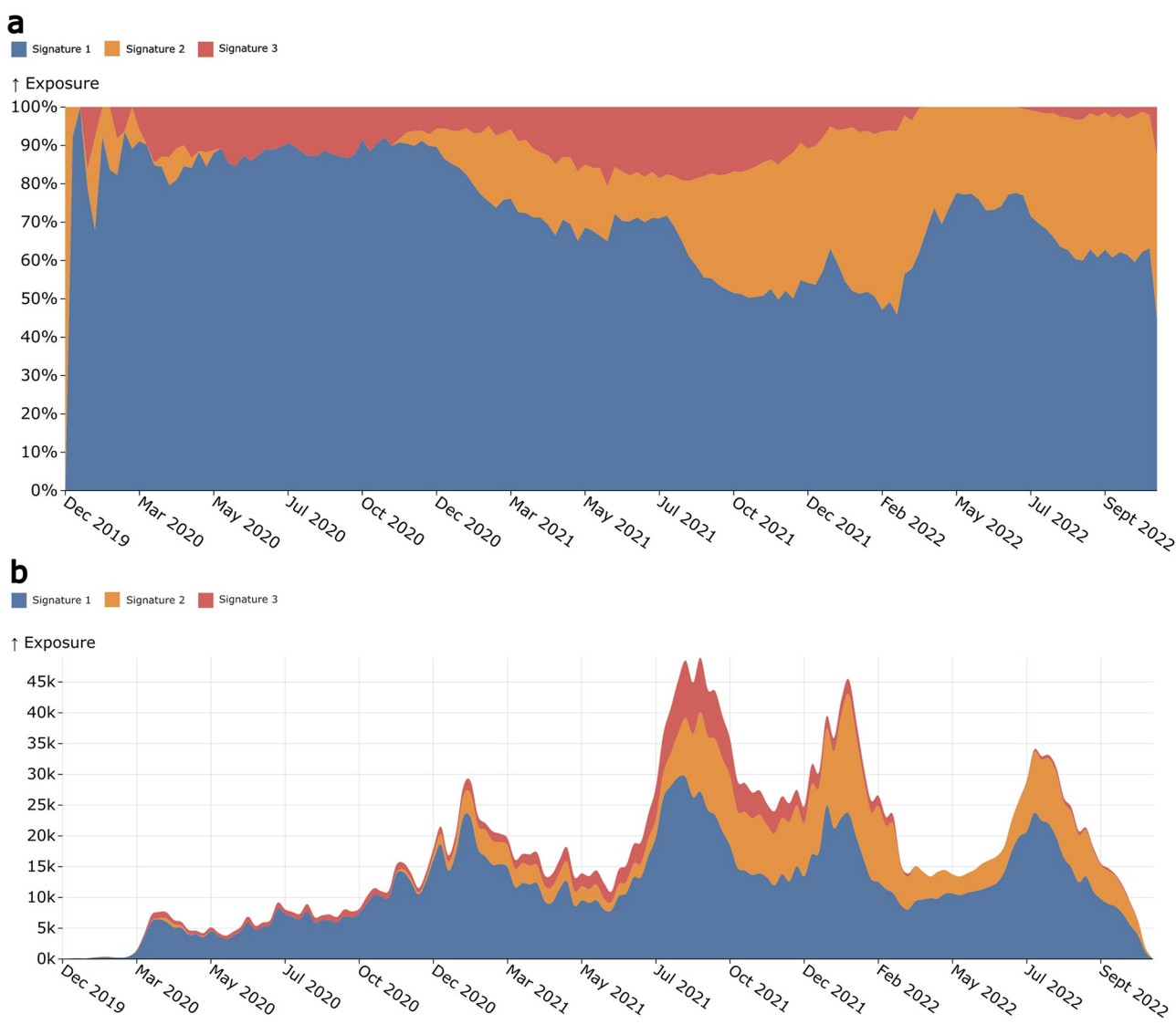

**Fig 4. Signature exposure plots showing the activities of the extracted mutation signatures over the duration of the COVID-19 pandemic. A**. Shows the percentage activity of the signatures during a given week of the pandemic, with each colour representing a different signature. **B**. Shows the signature activities as their absolute values at each epidemic week.

also observed by Bloom et al. [23] and Ruis et al. [24]. This is different to Delta, where there was an increase in Signature 3 following its emergence. These Signature 3 changes become particularly apparent when we begin to look at signature activities within variant-defined subsets of the data.

## Signature dynamics spatially and by variant

After observing changes in signature activity during transitions between dominant variants, we next investigated the differences between signature activities in variant-defined subsets of the data as well as in continent-defined subsets. We used the globally extracted signatures to extract exposures from the subsets using a non-negative least squares regression to retain the non-negativity constraint. This allowed for the measurement of signature activity in each of the subsets of interest.

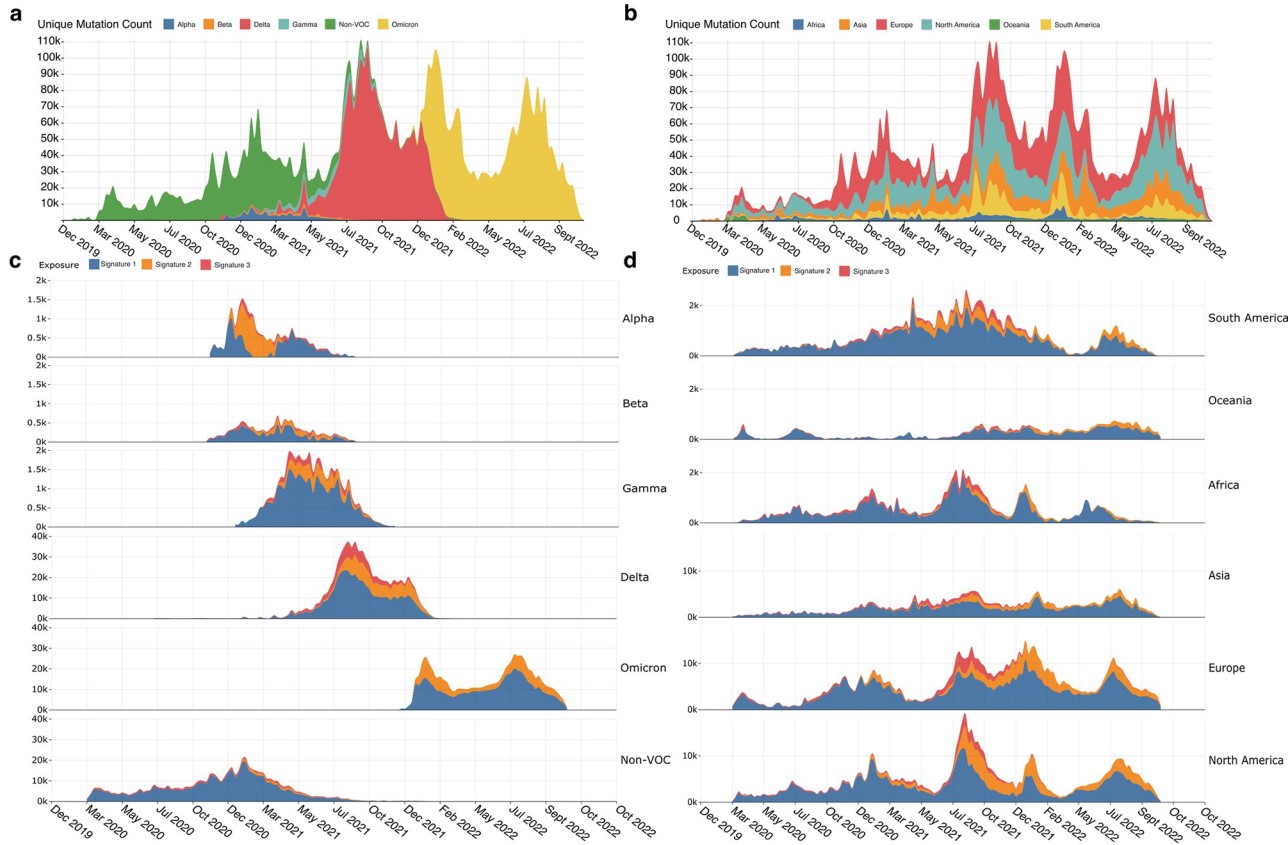

**Fig 5. A**. Counts of unique SARS-CoV-2 mutations for each epidemic week, with colours representing which continent the mutations came from. **B**. Counts of unique mutations per week that are part of the mutational signature substitution-context features (i.e., no indel mutations included). Colours represent which lineage/group of lineages the mutations belong to. **C**. Ridgeline plot showing the exposure of mutational signatures in SARS-CoV-2 variant-defined subsets. Exposures are coloured by the signature they have been attributed to. **D**. Ridgeline plot showing the exposure of mutational signatures in SARS-CoV-2 continent-defined subsets.

Signature 1 was the most active in almost all the variant-defined subsets as was expected from the global activity. Signature 3 was most active in the Delta subset as well as during the Delta wave in the continent-defined subsets (Fig 5). The non-VOC, Beta and Omicron subsets appear to be the least impacted by Signature 3 with almost zero activity in Omicron. Signature 2 also shows low activity in the non-VOC subset but is very active in the other VOC subsets, in particular Alpha, where it appears to be the most active, overtaking the Signature 1 process.

Continent-defined subsets of the data also consistently showed the high activity of Signature 1. Signature 2 begins to consistently appear in all continents after 2020, with only small bursts of activity being detected before this (Fig 5D), again consistent with what we see in the global data. Signature 3 activity also follows the pattern of the global activity, appearing most prominently during the Delta wave.

## Bridging the gap between mutation signatures and amino acid substitutions

Stratifying non-synonymous nucleotide substitutions by their association with mutational signatures should provide insights into how these mutational processes affect viral proteins. Exposures were calculated by stratifying nucleotide mutations by whether they were synonymous or non-synonymous substitutions for each dataset (Fig 6A). The unattributed exposure

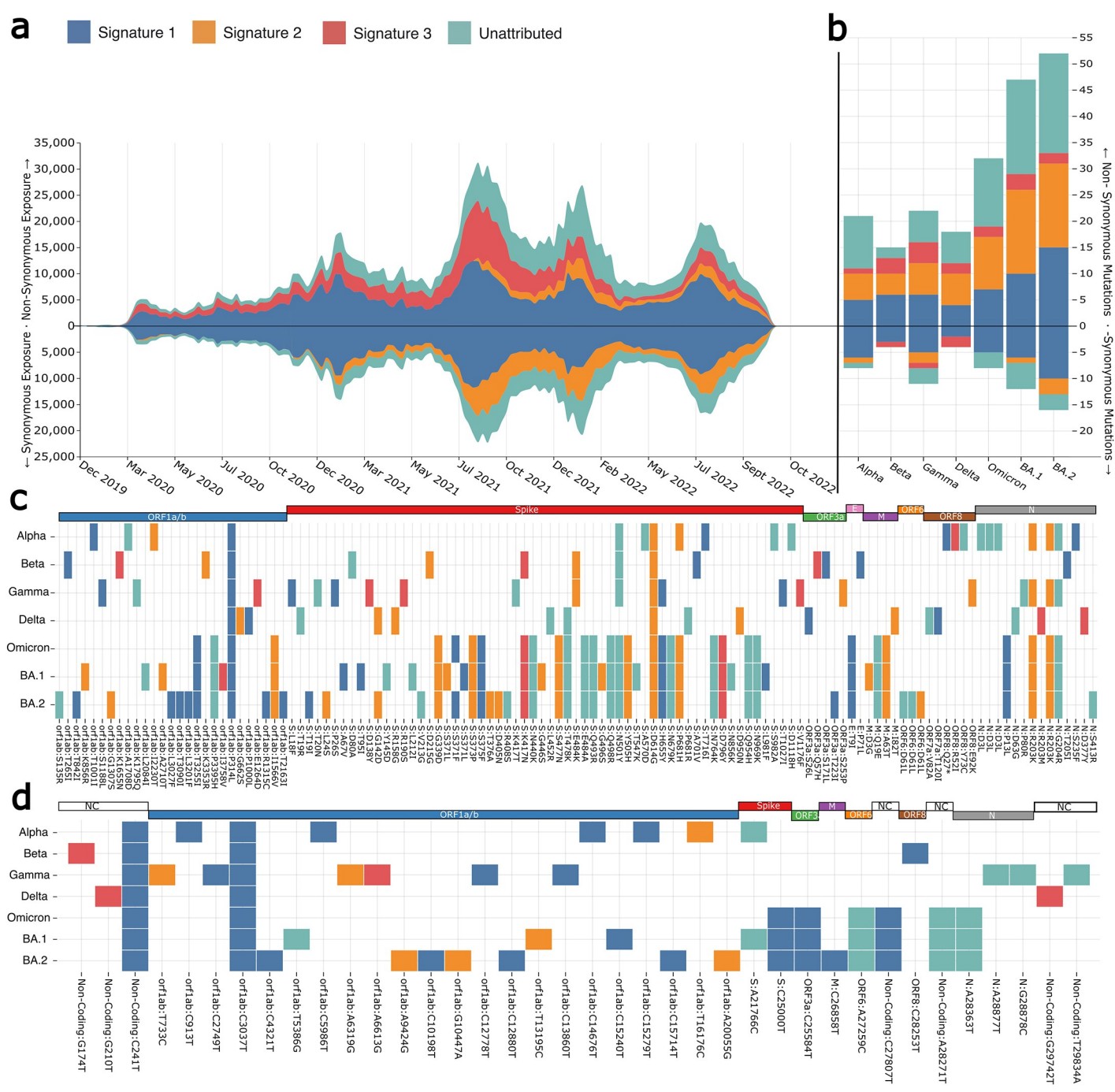

**Fig 6. A**. Exposures for each of the SARS-CoV-2 mutational signatures for both synonymous and non-synonymous stratified datasets. Synonymous exposures are below 0 on the y-axis, while non-synonymous exposures are above 0. Each area represents signature exposures across epidemic weeks, with colours representing which signature the exposures are attributed to. **B**. Non-synonymous and synonymous mutations in the tree-based references of identified variants of concern. Signature 1 produces the majority of both synonymous and non-synonymous substitutions in all lineages. Signature 3 mutations are more often non-synonymous substitutions in the lineages of concern, with most lineages having few to no changes. Signature 2 non-synonymous mutations appear to have increased in the Omicron lineages (BA.1 and BA.2). **C**. Variant of concern associated non-synonymous mutations coloured by the mutational signature with the greatest likelihood of causing the change. **D**. Variant of concern synonymous mutations coloured by the putative mutational process that caused the change.

was calculated using the model error for mutational categories not contained within any of the extracted mutational signatures. The majority of non-synonymous substitutions can be described by the observed mutational signatures. Signature 1 likely produces most of the non-synonymous mutations, however, Signature 3 is an almost exclusively non-synonymous signature, with particularly high activity during the Delta wave of infections. Signature 2 appears to produce predominantly synonymous mutations.

Using the tree-based references, we can also look at individual lineage reference sequences to observe which mutational processes have probably produced their specific amino acid substitution set. The tree-based references were used since they are equivalent to a high-quality representative sequence and because many of the early real sequences contain sequencing errors. For each variant of concern, mutations were assigned to a signature by calculating the maximum likelihood of the mutation and its context being produced by each of the three extracted signatures. Using the trinucleotide context $C[C \rightarrow T]G$ as an example, the likelihood function is $P(C[C \rightarrow T]G \mid Signature)$, which corresponds to the probability bars for CT-CCT in the extracted signatures. Mutations that contained substitution-context pairs not found within any of the mutational signatures were labeled as "unattributed".

The Alpha VOC tree-based reference sequence contains eleven Signature 1 changes, six Signature 2 changes and a single Signature 3 change. Signature 1 changes account for 39% of all substitutions within the Alpha tree-reference sequence, with 75% of these mutations being non-synonymous substitutions. Signature 1 was frequently active prior to the Alpha VOC's emergence. The activity plots (Fig 4) show that this was the case for much of the pandemic, particularly prior to the Alpha's emergence around September 2020. It should be noted that while Signature 1 mutations are by far the most frequent, only one is found within the Spike protein (producing the S:T716I change). Signature 3 only had one change, which was non-synonymous appearing in ORF:8. Signature 2 mutations were non-synonymous substitutions 83% of the time, with three Spike mutations relating to the process including S:D614G, which is present within all known variants of concern.

The Beta VOC emerged around the same time as Alpha (Autumn 2020) and is defined by a smaller set of mutations. A greater proportion of Signature 1 mutations are non-synonymous substitutions in Beta (66%). Signature 2 mutations resulted in S:D215G and S:E484K, the latter reported to help the virus evade neutralising antibodies [25]. Signature 3 mutations most likely produced S:K417N in spike, which is also reported to aid in antibody evasion [25, 26] similar to S:E484K.

Gamma also emerged in Autumn 2020 and has 33 different defining substitutions. Signature 1 mutations account for 11 of these with 54% being non-synonymous. Four are present in Spike including S:L18F, S:P26S, S:H655Y and S:T1027I. Signature 2 mutations resulted in six amino acid substitutions, with only 75% of changes being non-synonymous. Three of the five mutations in non-synonymous substitutions occurred in Spike. Signature 3 mutations in the Gamma lineage were all non-synonymous except for a single synonymous substitution in ORF1a/b.

Delta was the first VOC to dominate worldwide and replace almost every other lineage in all regions. The initial Delta sequence (Pango lineage B.1.617.2) contains six Signature 1 mutations. 66% of these changes were non-synonymous and none occurred within Spike. Signature 2 mutations were all non-synonymous and displaced throughout the virus ORFs including ORF1a/b, S and M. Signature 3 mutations in Delta are found in non-coding regions and N, with the N mutations both being non-synonymous.

Omicron is the most recent VOC to emerge, quickly replacing Delta globally. Omicron differs from earlier VOCs with a much greater number of Spike mutations relative to the other ORFs. The first identified Omicron variant B.1.1.529 has 40 substitutions of which 32 are non-

synonymous changes. This is almost double that of Delta, which only had 18. Seven of these substitutions were Signature 1 changes, two were Signature 3 and ten were Signature 2 changes. There are four non-synonymous ORF1a/b mutations despite this ORF being substantially longer than SARS-CoV-2's other ORFs. Only one Spike substitution was synonymous out of the 21 total changes. This number is even greater when looking at the major Omicron variants BA.1 and BA.2. BA.1 had 31 non-synonymous substitutions in Spike alone while BA.2 had 28. Between these three Omicron variants, only two Spike substitutions are non-synonymous out of a total of 40. Nine of the 40 changes are from Signature 1, 2 are from Signature 3 and 12 are from Signature 2. This means 23/40 of the changes appear to come from these three mutational processes. 20 of the 40 substitutions observed in these variants were present in the receptor-binding domain (RBD) of Omicron, with nine of these changes thought to help Omicron evade the immune response or increase its transmissibility [27]. Of these beneficial RBD changes, three are potentially the result of Signature 1 activity, 9 are Signature 2 and one is from Signature 3. The high density of Signature 2 RBD amino acid changes in a variant that has emerged as Signature 2 exposure increased suggests that the mutational process behind Signature 2 may have contributed to the emergence of the Omicron variant.

### Signature exposures and highly mutated sequences in wastewater data

Similar trends over time in exposures are seen when the mutational signatures are applied to publicly available wastewater data. Although the trend is seen at a lower resolution than global data, Signature 1 and Signature 3 are gradually replaced by Signature 2 (Fig 7A). Although, Signature 2 is not quite as strong as in the global data (Fig 4). This suggests trends in mutational processes can be monitored using wastewater, not only sequencing of the infected population. Additionally, at time periods where a high level of virus diversity is expected, there are highly mutated sequences present in the wastewater (Fig 7C). This suggests cryptic sequences in wastewater may be used to observe potential upcoming variants, similar to how known sequences have been back-traced to particular buildings using wastewater [28].

As chronic SARS-CoV-2 infections are implicated as a major contributor to VOC evolution [29, 30], it may be possible to parse highly-mutated cryptic sequences of interest from chronic infections out of wastewater data in the interest of detecting potential VOCs. Unfortunately, this is problematic to deconvolve as sequencing data for immunocompromised and chronically infected individuals is sparse. When sequences from known chronic infections are examined, the distribution of mutation types is consistent with global data, with Signature 1 mutations dominating as expected for samples from January 2022 (Fig 7B). Although, due to the low number of chronic infections for comparison this result is not very conclusive, it does demonstrate how mutational patterns can be potentially detected in this type of data. Studying these types of infections, and underlying mutational processes, will be important to understand better the origins of the sets of mutations that contribute to the generation of VOCs.

## Discussion

In this study, we investigated SARS-CoV-2 lineage dynamics and identified temporal variables that are associated with increased numbers of infection cases. Both public health measures and virus properties were associated with the sequential waves of regional SARS-CoV-2 infections cases. These predictors have varying impact in different geographical locations. As more of the global population's immune system becomes sensitised to existing SARS-CoV-2 variants, either through previous infection or vaccination, the virus has and will continue to undergo changes that enable reinfections. The continued emergence of new variants is thus expected. In some regions, government stringency had limited significant impact on patterns of

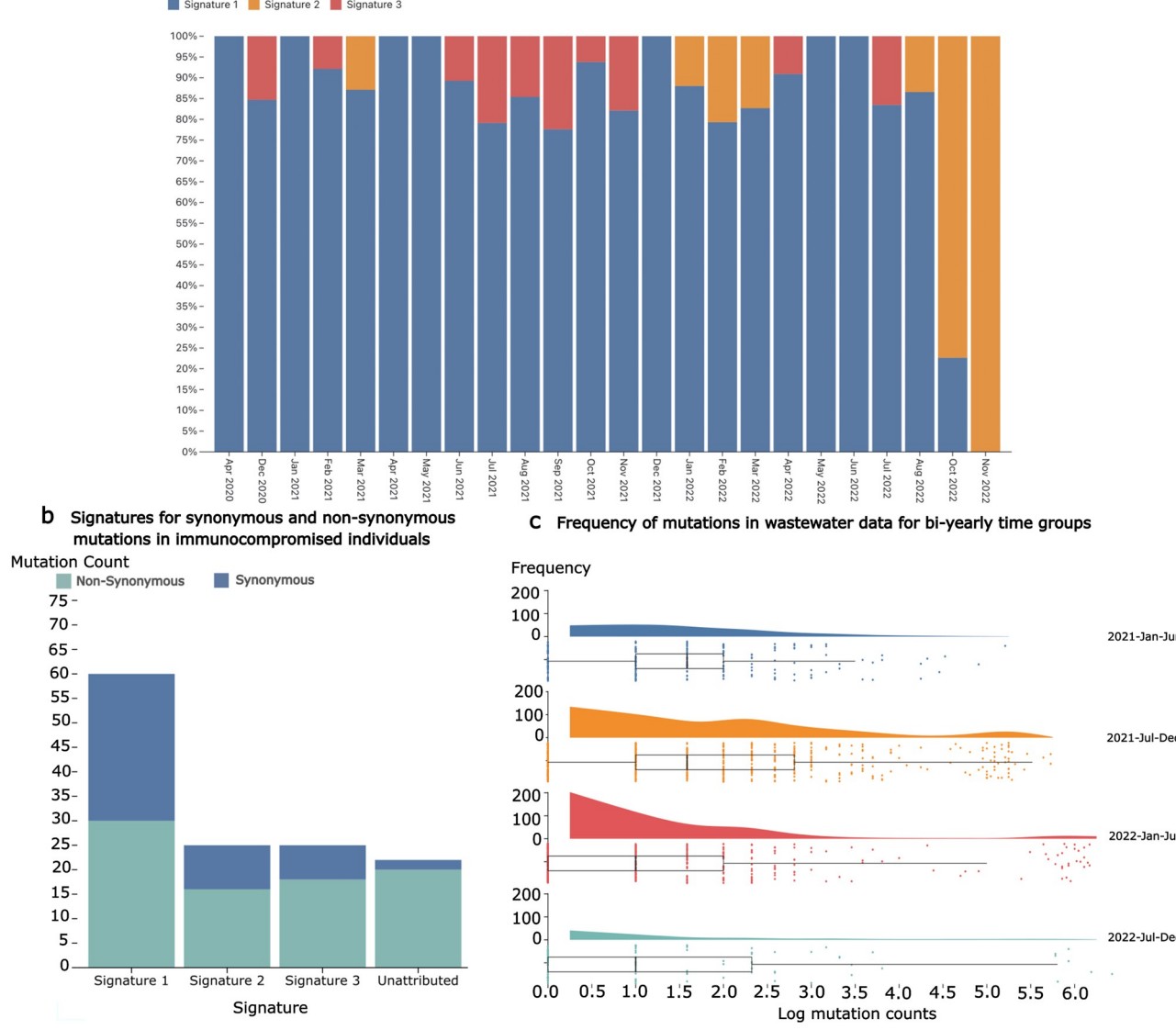

**Fig 7.** **A**. Signature exposures per month from wastewater sequences show similar trends in mutational processes as the global data, although at a lower resolution and, interestingly, with a lower Signature 2 exposure. **B**. Substitutions in SARS-CoV-2 consensus sequences from infections of immunocompromised individuals contain mutation types corresponding with patterns observed in the distinct signatures. Of note, there are more synonymous mutations present in the chronic infection data than in the global sequences, although it is important to note the sample size for immunocompromised infections is low. **C**. Mutation counts in wastewater sequences for bi-yearly time periods. Highly mutated sequences cluster to the right especially during the 2021 July-December time period, as would be expected when Omicron was emerging.

infection. This could be due to differences in implementation strategies and support, other competing predictor variables, as well as behavioural changes in citizens as a response to the restrictions.

Our analysis highlights the significant role of vaccination in influencing reported COVID-19 case patterns across all continents, even in regions with lower vaccination coverage like Africa. Despite Africa's lower vaccination rates, the continent has seen a relatively low-level of sustained transmission. This phenomenon might be attributed to factors such as the younger median age of the population, lower population density, immune priming due to prevalent

infectious diseases, and limited testing capacity [31]. The weak impact of viral diversity on reported cases in Asia and South America may be explained by the emergence and dominance of variants such Delta and Gamma in the regions, respectively. For instance, the Delta variant, initially identified in Asia, quickly became the predominant strain, overshadowing other lineages before spreading globally. Overall, the predictor variables significantly contributed to explaining the rise and fall of infection numbers across different continents, accounting for more than half of the variance in reported cases. The differences in the regression effectiveness can be attributed to intrinsic differences among continents, such as variations in vaccine coverage, testing and sequencing capabilities, and the effectiveness of government stringency measures.

While our model effectively captured the general trends of infection waves, it struggled to accurately represent peaks within short time-frames in some countries. This discrepancy might be attributed to the omission of certain predictor variables, like mass gatherings, which are known to contribute to viral super-spreading events [32].

In utilizing the OWID and OxCGRT datasets, which are arguably among the most comprehensive for addressing our research objectives, we note some limitations. First, there were discrepancies in parameter definitions, such as varying case classifications across regions. Second, positive tests are commonly labeled based on their reporting date rather than "date-of-event" [33]. Lastly, the cases reported in these datasets may not be fully representative of the actual disease burden. Although the Human Development Index (HDI) of a country can act as a proxy to bridge the gap between reported cases and the true disease burden, it does not fully capture the entire complexity.

The extracted signatures from the global SARS-CoV-2 dataset show clear and distinct patterns describing mutational processes acting on the viral genome. The most prominent of these signatures, Signature 1 (Fig 3 and S5 Fig), shows a marked bias towards C→T mutations, a signal indicative of the APOBEC family of cytidine deaminases [17, 18]. APOBEC enzymes have been shown to cause extensive C→T editing of DNA and RNA in human and viral genomes. However, it is not yet clear whether they are the cause of this pronounced C→T bias in SARS-CoV-2 despite a number of other studies also observing other APOBEC-like mutational patterns [34–37]. Cytosines flanked by either an adenine or thymine in both the 3' and 5' direction appear to be the most pronounced targets of Signature 1. APOBEC editing was shown to have contexts outside of the traditional TpC when structural features of the nucleic acid such as hairpin loops are present [38]. Outside of structural features, APOBEC3A is thought to be the predominant cause for TpC changes and is found to be expressed in lung tissue [39]. ApC changes are considered to be caused by APOBEC1, which in cell models was shown to efficiently edit SARS-CoV-2 RNA [39]. APOBEC1 is found predominately in the liver and small intestine, tissues reported to be infected by SARS-CoV-2 [39, 40]. 3' CpG nucleotide contexts are the most targeted, in particular TCG, CCG and ACG. CpG suppression is a well-known dinucleotide bias. In RNA viruses, this appears to be a result of selective pressures exerted from the presence of host CpG sensing molecules such as Zinc-finger Antiviral Protein (ZAP). ZAP relies on host CpG suppression to allow it to specifically target non-host genomic material (such as viral RNA) with higher CpG content [41]. This allows viruses with lower CpG content to better evade restriction by ZAP since it more closely resembles the host CpG composition. While ZAP does not induce C→T changes, it may help explain why C→T sites in a CpG 3' context are preferentially edited relative to other 3' contexts. ZAP has been shown to restrict SARS-CoV-2 despite pre-existing CpG depletion [42]. ZAP isoforms have been shown to prevent necessary translational frame-shifting for SARS-CoV-2 ORF1b protein production. [43]. The non-normalised form of Signature 1 (S5 Fig) shows that when tri-nucleotide bias is not accounted for 3' CpG's are lower than the normalised signatures, yet 5' TpC and

ApC contexts remain the most prevalent(S5 Fig). The most targeted contexts do shift to ACA, ACT and TCT, likely reflecting their comparatively high abundance within the SARS-CoV-2 genome relative to 3'CpG contexts. These non-normalised contexts are consistent with what was earlier reported by Simmonds et al. [17]).

Signature 2 (Fig 3 and S5 Fig) has a nearly identical proportion of A→G and T→C mutations. These are a known target of the ADAR family of adenine deaminases. ADAR enzymes typically operate on double-stranded RNA and convert adenine into inosine [21, 22]. Inosine forms base pairs with cytosine, which after another round of replication causes guanine to replace the inosine and complete the A→G change. As ADAR operates on both strands of dsRNA, the mutational signature resulting from the process is expected to contain an equal proportion of A→G and T→C mutations, which is the case for Signature 2 [21]. Signature 2 also contains a number of G→A mutations, which could be caused by low-level C→T activity on the negative sense RNA strand. Due to the cellular strand biases present between the positive and negative sense RNA [36], C→T mutational processes acting on ssRNA are much less likely to produce a mutation on the negative strand (resulting in G→A substitutions) than C→T changes on the positive strand. The negative strand will only be present during the replication phase of the virus while the positive strand will be present both on cell entry and on exit as the new viral particles are packaged to infect further cells. This could explain why the negative sense Signature 1 changes are present in Signature 2, since it may be operating at a similar level to Signature 2 on the negative strand. The non-normalised form of Signature 2 (S5 Fig) does have different targeted contexts, just as with Signature 1. However, the main attribute of Signature 2 is its equal contributions of A→G and T→C substitutions, which still remain equal.

Signature 3 (Fig 3 and S5 Fig) is dominated by G→T substitutions. A putative mechanism for this is Reactive Oxygen Species(ROS) in the cell. Increases in oxidative stress as part of a ROS 'burst' have been associated with viruses during the early stages of infection [34, 44]. Guanine nucleotides are known to be vulnerable to oxidation, with the product 7,8-dihydro-8-oxo-2'-deoxyguanine (oxoguanine) pairing with adenine bases rather than cytosine [44, 45]. Similar to inosine causing A→G changes, this change to oxoguanine will result in a G→T mutation after a replication cycle. The lack of C→A changes in the signature also suggests that the mechanism is most active on the positive single-stranded RNA rather than the negative single-stranded RNA. The initial positive single-stranded RNA is found in the cytoplasm, meaning it can be easily accessed by ROS and other mechanisms of mutation. Viral replication is thought to take place within membrane-bound environments that aim to protect the RNA. The presence of double-stranded RNA within these environments strongly suggests that this is the case [46] and may explain the relative lack of negative strand mutations in SARS-CoV-2 signatures. The non-normalised G→T signature (S5 Fig) seems to display a context preference of TpG and ApG nucleotides, although this contextual bias is changed to CpG and ApG following normalisation. These contextual biases mean that the signature could be some other as yet unknown editing mechanism on the viral RNA, although normalisation changing this context so heavily suggests that this bias perhaps has more to do with genome composition. The increased CpG context shift post-normalisation could also be another ZAP-induced effect, where CpG depletion is selected for to help the virus evade ZAP. Curiously, this G→T bias has been observed in other coronaviruses, but not widely among RNA viruses [47]. ROS has a verified cancer mutational signature [15, 48] although the context preferences do not match the signatures (normalised or non-normalised) observed here. However, there are a multitude of differences between viral RNA and human DNA that make these signatures difficult to compare.

It is important to note that while SARS-CoV-2 does have an error correction mechanism resulting in fewer replicase-induced errors, this mechanism will not catch all changes. A

number of the mutations picked up from the set of sequences (and included in our mutational signatures) will be derived from replication errors. However, the clear and repeatable extraction of the signatures indicates that despite this potential contamination, the extracted signatures do appear to be predominantly other mutational processes. While a replication error-associated mutational signature may be identified in future, this signature is too diffuse to identify as a distinct process. Similarly, a high proportion of mutations are not accounted for by the extracted mutational signatures. These mutations were not present in large enough quantities to enable effective extraction from the data. Future methods may be able to tease out the more subtle mutational mechanisms that almost certainly exist to induce these less common mutation types.

Signature activities clearly change in both the global dataset and in the various subsets of the data for VOCs and continents. In the global data (Fig 4) Signature 1 is dominant throughout the pandemic. Signature 2 only begins to appear around November 2020, after which it appears consistently active for the remainder of the pandemic. This is approximately when variant of concern lineages began to emerge, as well as the beginning of the first vaccine rollouts. This is particularly apparent in the Alpha subset where Signature 2 is the most highly active mutational process (Fig 5), with a large depletion of Signature 1 activity as well.

Alpha was shown to increase sub-genomic RNA expression of several immune-antagonist viral proteins including nucleocapsid (N), ORF9b and ORF6 [49–52]. N is thought to shield dsRNA from detection by RNA sensors, which trigger downstream antiviral response pathways [49, 52–54]. ORF9b antagonises TOM70, a protein required for the activation of mitochondrial antiviral-signalling proteins (MAVS) [49] while ORF6 inhibits the transportation to the nucleus of inflammatory transcription factors [55]. Combined, the cumulative immune inhibition may have resulted in an observable change in the mutational processes that we observe within the Alpha lineage. Beta and Gamma (both VOCs that emerged around the same time as Alpha) gained amino acid substitutions that helped evade the immune system primarily via antigenic change. Alpha's reliance on attenuating immune pathways rather than antibody binding may be why we see a different signature exposure pattern in this VOC relative to the others. This could be due to the attenuated pathways being involved in signalling for the mutational processes behind Signatures 1 and 3, while not inhibiting Signature 2 as much.

This Alpha pattern is not observed in the other VOC datasets, although Delta and Omicron have a high level of Signature 2 exposure as well, despite Signature 1 remaining the dominant process in those subsets. Signature 3 appears to be most prominently found in the Delta subset and remains consistently at low levels in the global data until January 2022 when it appears to disappear almost entirely. The Omicron subset has little to no exposure for Signature 3 and this happens to be the VOC almost exclusively circulating after January 2022. Why Omicron appears to have so little Signature 3 exposure is unclear, although unlike previous VOCs, Omicron differs in its preference of cell entry mechanism. Previous variants of the virus typically enter the cell using membrane fusion, where the viral membrane fuses with the cell membrane via the action of ACE-2 receptor binding and TMPRSS2 cleavage of the spike protein. Omicron instead favours an endosomal route of entry whereby the viral particle binds to the cell using ACE-2 and is enveloped by endocytosis into the cell. Cleavage of the spike protein then occurs via the action of Cathepsin L, which allows for the release of the viral RNA into the cytoplasm of the now-infected cell [56, 57].

Signature transitions from Signature 1 to Signature 2 changes occur from December 2020 onwards in the global dataset and appears consistently in the VOC and continent-defined subsets around this time point as well. Alpha underwent a major shift to Signature 2 mutations early in its time as a VOC, although Signature 1 returned as the predominant set of changes towards the end of its wave of infections. The non-VOC subset appears to be the least impacted

by Signature 2 changes. However, this can mostly be explained by the number of non-VOC sequences quickly declining after the emergence of the VOC lineages. Delta underwent a dramatic increase in Signature 2 and Signature 3 exposure from July 2021, with Signature 2 becoming the predominant signature towards the end of Deltas wave. Signature 2 changes continue into Omicrons introduction, although it does decrease after the initial BA.1 wave from December 2021 to March 2022. It seems clear that while Signature 1 mutations have dominated in contributing to the evolutionary capacity of SARS-CoV-2 throughout the pandemic, this mutational environment is beginning to change. Such shifts in mutational processes are potentially evidence of changing interactions between the viruses and the immune systems of the hosts they circulate within. For example, changes in population-level immunity via vaccination or previous infections may influence the mutations that we observe in the data. Changing mutational process activity in consensus sequences from infections is unlikely to fully reflect the true activity of each process, but they are likely to show which processes are contributing mutations that eventually make it into circulating viruses.

All variants of concern we assessed show predominantly non-synonymous mutations and all mutational signatures are associated with more non-synonymous than synonymous changes. More synonymous substitutions in the lineage references were found in ORF1a/b, which is expected due to it being the longest ORF. However, this pattern is not observed with non-synonymous mutations as these are mainly located in the spike protein (Fig 6C and 6D). This is consistent with spike being under intense immune pressure since it is the main glycoprotein for SARS-CoV-2. As such, spike must change in order to escape the host immune response, while maintaining its main function of binding and entry into host cells. Signature 1 changes are the predominant source of mutations in all SARS-CoV-2 VOCs that we analysed, followed by unattributed mutations, Signature 2 changes and Signature 3 changes. Signature 3 changes were unlikely to be synonymous mutations with only Beta, Gamma and Delta containing very few such changes (Fig 6D). This is also reflected in the global synonymous/non-synonymous exposures where Signature 3 appears completely inactive in the synonymous mutation subset (Fig 6A). Signature 2 exposure appears the most likely to be synonymous mutations (Fig 6A) but this does not seem to be observed in the VOC lineages where most Signature 2 changes are non-synonymous mutations (Fig 6B).

In conclusion, mutational signature analysis reveals important processes contributing to SARS-CoV-2 genetic variation and serves as a tool to track the dominant changes over time and to generate hypotheses about the main mechanistic processes in play. Specifically, host antiviral molecules as opposed to replication errors appear to be a the main generator of mutations (confirming earlier computational studies), a result that requires experimental confirmation. Despite limitations in potential biases, our findings contribute to a better understanding of the complex dynamics driving the evolution of SARS-CoV-2 and the emergence of VOCs.

## Methods

### Data

The findings of this study are based on metadata associated with 13,281,213 sequences available on GISAID up to October 26, 2022 and accessible at doi.org/10.55876/gis8.221201qs. Sequences were filtered to remove records from non-human hosts, with lengths less than 20,000 nucleotides, non-assigned lineages, with greater than 30% unknown bases, sequences reported to be collected before 24/12/2019 and those with excessive mutations/deletions. The cutoff for filtering out hypermutated sequences was 175 mutations in coding regions or more than 69 different deletions, the cutoffs were manually determined after evaluation of the

mutation/deletion distribution and selecting the point where sequence counts were consistently observed in single digits, this resulted in 1,852 sequences being filtered out.

Publicly available daily SARS-CoV-2 cases, tests performed and total vaccinations per capita were obtained from OWID [58] in September 2022. Prior to February 2023, the OWID data was piped from the Johns Hopkins University COVID-19 dashboard [33, 59]. Country-level government stringency indices were downloaded from OxCGRT [60]. Government stringency indices are composed of nine indicators: school closure, workplace closure, cancellation of public events, stay at home order, public information campaigns, restrictions on public gatherings, public transport, internal movement and international travel. The index on a given day ranges from 0 to 100 and is calculated as the mean of the nine indicators, with higher indices indicating stricter regulations. If responses vary at sub-national levels, the index at the strictest level is used [60].

Wastewater findings are based on metadata associated with 1,343 sequences available on GISAID and accessible at doi.org/10.55876/gis8.230406qg. Wastewater sequences were downloaded from the 'wastewater data' section of GISAID in December 2022.

Sequences for immunocompromised individuals were downloaded from GISAID in November 2022. Analysis of this was based on the metadata associated with 34 sequences available on GISAID and accessible at doi.org/10.55876/gis8.230406fb. Sequences were chosen based on the known list of sequences used in [30]. Sequences were aligned to the COVID reference genome before use.

## Design

Predictors of SARS-CoV-2 reported cases were explored using a linear model at both country and continent levels. We collected continuous dependent variables reported on a daily basis. These were classified into two groups: (i) public health measures (government stringency, testing capacity and vaccination), (ii) viral properties (diversity and fitness). We examined the data for completeness of predictive variables. In instances of missing vaccination data, we interpreted this as no vaccinations having been given. This was a reasonable assumption for periods prior to the vaccine rollouts in the respective countries. With the exception of vaccinations, variables with less than 70% of the countries reporting data were not included. The number of SARS-CoV-2 diagnostic tests performed was excluded as a predictor due to missing data. We determined the previous burden by summing the adjusted new cases per capita over the past 90 days. Prior infection significantly reduces the risk of a subsequent infection, with a reduction in risk of up to 95% in the initial three months [61]. This was included as a predictor variable in the linear model.

Amino acid substitutions were defined against the Wuhan-Hu-1 sequence. Building on findings from Obermeyer et al., we extracted a list of previously identified fitness-associated mutations [62]. Each fit mutation within a sequence was counted and the counts were normalized to the number of sequences per geographical location. Virus fitness was therefore defined as the sum of the frequencies of previously identified [62] amino acid substitutions that increase SARS-CoV-2 fitness divided by the sum of total genomes and the log of total mutations per location.

$$Virus\ Fitness = \frac{weekly\_sum\_of\_fit\_mutations}{total\_seqs\_per\_week\ +\ log(total\_mutations\_per\_week)}$$

Diversity was calculated by dividing distinct lineages by the total number of genomes in a given week. Sequences reported in GISAID were assumed to be representative of the diversity of infections for that continent/country.

## Linear model

We employed a linear regression model, described by Heo et al. [63], to adjust reported cases per country using the Human Development Index (HDI), which encompasses not just economic growth but also reflects a country's capacity for per capita testing. Countries with higher HDI levels, typically high-income nations, conducted more tests per million people, often leading to more confirmed cases compared to nations with lower HDI levels. Adjusted daily cases were smoothed using a 14-days rolling average to limit possible noise and identify simplified changes over time. For continent-level analysis, data from all contributing countries was used to fit the linear model. To ensure that countries with a large number of cases didn't artificially inflate the results, each country's influence on the continent-level OxCGRT index was adjusted based on its percent contribution to the continent's 14-day average daily case tally.

Pearson's correlation was used to test for correlation among the variables. Multiple linear regression was fitted to evaluate the relationship between infection rate (adjusted daily cases per capita) as the outcome and the public health measures and viral properties as predictors within the different continents. The regression models were fitted on data from 01 April 2020 onwards, as (sequence) data addition remained stable after this. The country-level analysis was carried out for countries with less than 50 days of missing genome data using a similar approach.

## Pandemic plots

Case numbers and sequence data were aggregated by their respective continents, a 14-day rolling average was used to smooth out daily infection rates and categorical variables were summarised by counts. Proportions of lineages were calculated in 14-days bins and the most common lineages were visualised per continent.

## Tree-based referencing

The rapid evolution of SARS-CoV-2 means that the majority of viral sequences are distinct from the early pandemic reference genome Wuhan-Hu-1 [64]. Continuing to count mutations against the early reference sequence can result in mutations being allocated the wrong substitution category (i.e., A→T instead of a C→T) where sites have mutated multiple times. Azgari et al. [35] tackled this issue by building a tree of clustered sequences to remove ancestral mutations. However, we utilise the available SARS-CoV-2 tree generated as part of the Pango [8] nomenclature to generate a reference sequence for each defined lineage. This means that sequences from the lineage B.1 are compared against a generated reference sequence for the B lineage rather than the Wuhan-1 sequence (See S3 Fig for diagrammatic description).

One reference sequence was generated for each of the Pango lineages in the alignment. A nucleotide was included in the generated Pango reference if it exceeded a frequency threshold of greater than 75% of the samples from the lineage. If this threshold was not reached, the reference nucleotide of the nearest parental lineage was used (i.e., if a mutation in B.1 is ambiguous, the nucleotide from the B lineage reference at that position is used). Building intermediate references also meant that counting inherited mutations could be avoided. Since mutations were identified relative to their nearest parental Pango lineage, inherited mutations are not counted because, relative to this sequence, there hasn't been a mutation. Mutations are also only counted once per lineage set of sequences so that mutations that are observed many times due spread of the virus rather than acquisition by a mutational process are not over-counted. This means that convergent amino acid substitutions can be observed between lineage sets, although they may be undercounted within a lineage. However, this is necessary since it is very difficult to identify convergence within similar sequences (especially at a global scale).

Overcounting of the mutations results in mutational signatures that reflect the circulating predominant lineages rather than the mutational processes producing the mutations in those lineages.

## Pseudo-sampling

Mutations were binned into categories composed of their substitution type (e.g., cytosine → thymine = CT) and their mutation context. The mutation context is the mutated base and the nucleotides at the 5' and 3' positions of the mutated base. There are a total of 192 types of substitution-context matchings that can appear (12 possible single nucleotide changes x four possible nucleotide 5' x four possible nucleotide 3'). Every sequence produces a single count vector of mutation category counts, with the total count matrix becoming the mutational catalogue of the virus. On average, a single SARS-CoV-2 genome sequence has very few new mutations. As extracting mutational signatures when mutation counts are low is unlikely to produce meaningful results, we define each sample as a time-point (all of the sequences collected in an epidemic week) and decompose signatures from the counts at each time-point rather than from each sequence. This shrinks the mutational catalogue of the virus from millions of samples down to less than 200 samples, one for each Epidemic Week.

## Non-negative matrix factorisation

NMF (non-negative matrix factorisation) [65, 66] was used to split the mutational catalogue into two sub-matrices. One matrix represents the mutational signatures, the other matrix represents the exposure of the signatures. These matrices were used to reconstruct the original mutational catalogue with some degree of error. To verify the validity of the identified signatures, NMF was performed 100 times for each value of N, with N representing the number of signatures to extract from the mutational catalogue. For this analysis, N was set to 2, . . ., 10. For each NMF run, a new mutational catalogue was generated using bootstrap re-sampling of the original matrix and removal of any mutational categories that did not account for more than 0.5% of mutations. Mutational categories are pseudo-sampled down into epidemic week matrices that NMF was run on. The signatures were then clustered together using K-means clustering, with the cluster means forming the new signatures. Clusters were then assessed using the silhouette score to determine the clustering quality. Clusters with high silhouette scores are well separated from other clusters and are dense and well-formed. Cosine similarity was used to determine if the signature was reliably extracted from the cluster. The cosine similarity was calculated between signatures extracted from the whole mutational catalogue and the cluster means of the signature clusters. A higher cosine similarity indicates that the cluster mean shows a similar pattern to the initial mutational signature. Following the best practices in Islam et al. [66], an N value of three was selected due to the reduction of the reconstruction error plateauing around three and the marked decrease in silhouette score for signatures greater than 3. The average cosine similarity between signatures and clusters was consistently above 0.95 for each cluster and had an average of 0.98 for all three clusters when clustering was repeated 100 times. Silhouette scores for each cluster were above 0.95, suggesting excellent separation and density of clusters (S5 Table and S9 Fig). Signatures can therefore be reliably extracted from the bootstrapped catalogues, are robust and thus are unlikely to be artefacts. Counts of mutations were normalised by the tri-mer composition of the SARS-CoV-2 reference sequence (dividing the counts by the number of contexts in the reference sequence). Composition biased versions of the signatures were then produced by rescaling the signatures using tri-mer composition.

### Non-negative least squares regression

A non-negative least squares (NNLS) Regression was used to produce positive exposure weights for each of the signatures in each of the datasets. The non-negativity of the regression ensures that the weights of the signatures continue to represent an additive process. The NNLS weights can then represent the exposures of the signatures on each dataset.

### Consensus lineage and continent signatures

Mutational catalogues were constructed for each continent and each of the Variant of Concern (VOC) lineages (Alpha, Beta, Gamma, Delta and Omicron). The global signatures were then used to extract exposures for each of the mutational catalogues to determine how processes varied between each mutational catalogue subset. VOC sequence sets were filtered so that weeks with fewer than 100 sequences were excluded.

## Supporting information

**S1 Fig. Country-level SARS-CoV-2 lineage dynamics.** Solid bars show the biweekly proportions of the common lineages. Bars are coloured by lineage and white space shows the proportion of sequences from other lineages. The countries included in this analysis is based on temporal data completeness.
(TIF)

**S2 Fig. Model-fitting of country-level SARS-CoV-2 reported cases.** Black solid lines show a 14-day rolling average of adjusted SARS-CoV-2 cases. Pink solid lines show fitted mean response values of infection rates with predictor values as input and grey shaded areas highlight the confidence intervals. The countries included in this analysis is based on temporal data completeness.
(TIF)

**S3 Fig. Diagrammatic depiction of how tree-based referencing works.** Each Pango lineage has a reference generated for it. Arrows show which sequences use which reference sequence, with the arrow tip indicating the reference. For example, sequences from the B.1 lineage are compared against the reference for the B lineage so that B.1 lineage-defining mutations can be counted.
(TIF)

**S4 Fig. Graphical description of the methods for NMF extraction of mutational signatures.** For every value of N signatures, the mutational signatures are extracted 100 times for bootstraped and pseudo-sampled datasets. Once this has been completed, signatures are clustered into N clusters and the stability and density of those clusters are evaluated using the silhouette score. Signatures that have silhouette scores above 0.95 are evaluated as stable signatures. The cluster means become the extracted signatures. The best set of N signatures is selected by picking the value of N that best minimises the reconstruction error and has the best silhouette score (with a minimum of 0.95). A further evaluation is the cosine similarity of the clustered signature means with the signatures extracted by completing NMF on the original pseudo-sampled dataset. Again, signatures must have a cosine similarity of at least 0.95 to be considered.
(TIF)

**S5 Fig. Non-normalised mutational signatures for SARS-CoV-2.** Signatures were extracted using normalised counts calculated by dividing the mutation counts by the count of the tri-nucleotide context of the mutation context (Fig 4). These signatures were then multiplied post-analysis by the tri-nucleotide composition of the reference sequence to produce the non-

normalised signatures shown here.
(TIF)

**S6 Fig. Counts of unique substitutions per week of the pandemic.** Areas are coloured by substitution category.
(TIF)

**S7 Fig. Counts of unique substitutions per week of the pandemic for each VOC category.** Areas are coloured by substitution category.
(TIF)

**S8 Fig. Counts of unique substitutions per week of the pandemic for each continent category.** Areas are coloured by substitution category.
(TIF)

**S9 Fig. Signature evaluation metrics.** The number of signatures was selected at N = 3 since this produced an "elbow" for the reconstruction error while having a suitable silhouette score greater than 0.95.
(TIF)

**S1 Table. Proportion of common lineages/variants globally.**
(XLSX)

**S2 Table. Correlation between infection rate and predictor variables across different continents.**
(XLSX)

**S3 Table. Effect of public health measures (government stringency and vaccination) and viral properties (diversity and fitness) on infection rates at continent level.**
(XLSX)

**S4 Table. Effect of public health measures (government stringency and vaccination) and viral properties (diversity and fitness) on infection rates at national levels.**
(XLSX)

**S5 Table. Evaluation Results for Signature with N = 3.**
(XLSX)

## Acknowledgments

We gratefully acknowledge all data contributors, i.e., the authors and their originating laboratories responsible for obtaining the specimens and their submitting laboratories for generating the genetic sequence and metadata and sharing via the GISAID Initiative, on which this research is based. For the purpose of open access, the author has applied a Creative Commons Attribution (CC BY) licence to any Author Accepted Manuscript version arising. We thank Spyros Lytras, Francesca Young, Sejal Modha, Andres Gomez and Procheta Sen for their helpful comments throughout the process of writing and preparing this manuscript.

## Author Contributions

**Conceptualization:** Kieran D. Lamb, Ke Yuan, David L. Robertson.

**Data curation:** Richard J. Orton.

**Formal analysis:** Kieran D. Lamb, Martha M. Luka, Megan Saathoff.

**Funding acquisition:** Matthew Cotten, Ke Yuan, David L. Robertson.

**Investigation:** Kieran D. Lamb, Martha M. Luka, Ke Yuan, David L. Robertson.

**Methodology:** Kieran D. Lamb, Ke Yuan.

**Resources:** Kieran D. Lamb.

**Software:** Kieran D. Lamb.

**Supervision:** My V. T. Phan, Matthew Cotten, Ke Yuan, David L. Robertson.

**Visualization:** Kieran D. Lamb.

**Writing – original draft:** Kieran D. Lamb, Martha M. Luka, Megan Saathoff.

**Writing – review & editing:** Kieran D. Lamb, Martha M. Luka, My V. T. Phan, Matthew Cotten, Ke Yuan, David L. Robertson.

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
