## [Decision Letter · Decision Letter 0]

23 Aug 2023

Dear Prof. Robertson,

Thank you very much for submitting your manuscript "SARS-CoV-2’s evolutionary capacity is mostly driven by host antiviral molecules" for consideration at PLOS Computational Biology.

As with all papers reviewed by the journal, your manuscript was reviewed by members of the editorial board and by several independent reviewers. In light of the reviews (below this email), we would like to invite the resubmission of a significantly-revised version that takes into account the reviewers' comments.

We cannot make any decision about publication until we have seen the revised manuscript and your response to the reviewers' comments. Your revised manuscript is also likely to be sent to reviewers for further evaluation.

Sincerely,

Roger Dimitri Kouyos

Academic Editor

PLOS Computational Biology

Amber Smith

Section Editor

PLOS Computational Biology

Reviewer's Responses to Questions

**Comments to the Authors:**

Reviewer #1: I very much enjoyed reading this manuscript as a preprint and am grateful to the authors and editors for giving me the opportunity to perform peer review. In this manuscript, Lamb et al pprovide a tour de force summary of the mutational signatures experienced by SARS-CoV-2 over the course of the pandemic. The authors isolate three signatures, which are predominantly characterized by APOBEC-like, ADAR-like, and ROS-like substitution patterns. The results are compelling, well-described, and consistent with and make a substantial contribution to a growing body of literature recognizing the potential for substitutions in RNA viruses to be driven by processes divorced from selection of polymerase errors.

Major Comments

Section 2.2 (“covariates of the wave”) tries to tackle a very difficult question of what “drives” a SARS-CoV-2 wave. This isn’t the key result of the manuscript. This question requires exceptionally careful analysis to answer effectively. In general, the results and its associated methods section are somewhat underdeveloped and do not meet this requirement. This weakness doesn’t detract from the rest of the manuscript. That being said, an entire manuscript could be devoted to answering this question and I do not think it is appropriate that the authors change the scope of the manuscript. Instead, I would like to see some small changes to the analysis to increase the robustness of the results and slight dampening of the language in the results and discussion about the impact of this section.

- Both the OWID dataset and the OxCGRT datasets are likely the best datasets to answer the questions posed by the authors but limitations of each dataset should be mentioned in brief in the discussion. For OWID, I would especially call attention to the limitations of “date of report” based time series (see Dong et al. 2022 in Lancet Infectious Diseases for discussion of this).

- The regression of daily cases to government stringency index needs to be adjusted for the latter metric to be informative. The continent-level 14-day moving average is an appropriate means of dealing with the noisiness of reported case data, where there are frequent gaps in reporting that appear to be zero daily cases. The inappropriate step in this analysis is comparing the continent-level 14-day moving average of new cases to the average OxCGRT index of the component countries within the continent. The daily case rate will naturally be biased toward the countries with the highest case numbers (who contribute the greatest proportion of cases to the average) while the average OxCGRT index will have equal contributions from each country. In a scenario where only a handful of countries have low stringencies indices and these same countries are responsible for an outsized number of cases, any positive effect of government stringency will be dampened (as the OxCGRT index will effectively be artificially inflated). I highly recommend repeating this analysis but weighing a country’s contribution to the continent-level OxCGRT based on its contribution to the continent-level 14-day average daily cases within the same time window. From my assumptions of how the analysis scripts have been written, I don’t anticipate this being too onerous of a calculation. However, if this is challenging other approaches could be considered. I recommend this approach rather than weighing by population, as the former will still partially control for differences in population (India will be weighed more heavily than Nepal, for instance) while the latter will have the issue of more populous countries not necessarily having infrastructure for widespread community testing.

- This does not need to be adjusted by the authors, but for the sake of documentation part of the “careful analysis” referenced above would include empirical testing of the assumption “Sequences reported in GISAID were assumed to be representative of the diversity of infections for that continent/country.” Performing these analyses is not key to the interpretability of the later results section and for that reason I don’t consider it necessary.

The authors make a very astute observation on page 18 that the ROS-like signature appears to have context preference: “The G→T signature itself seems to display a context preference of TpG and ApG nucleotides”. This observation can (and should) be tested quantitatively. The same approach as in PMID: 34061905 can be used to calculate normalized context preferences. As one will need to compare to a reference sequence, I would approach this by calculating ratios for each tree-based reference for the pangolin lineages (using the observed substitutions for those references) and then calculating a median ratio from all of the values. This way, you’ll be able to robustly determine whether or not any contexts are over- or underrepresented. Dinucleotide context is probably sufficient, but if it is computationally straighforward then trinucleotide context would also be useful and interesting. The reason I view this as key to the manuscript is that the observation of a context preference for ROS would either (a) indicate context and structural preference for ROS, which would be a novel finding or (b) bolster the argument that the G—>T phenotype may be due to another as of yet defined nucleic acid editing mechanism. Either of those findings would be highly significant to the overall story and for that reason deserve robust statistical analysis.

In pages 17 and 18, the discussion of potential molecular mechanisms underlying the (lack of) ROS-like signatures in Omicron is excellent. I am very interested in what hypotheses and explanations the authors are able to generate for the preponderance of the ADAR-like signature in Alpha. The data in figure 5 is particularly striking and I think this result deserves more attention in the discussion that it is currently given.

On a smaller (but important) note, additional clarification is needed on page 20 in the methods to ensure reproducibility. Please state on which date the daily case data was downloaded from Our World in Data’s GitHub repository.

If prior to February 20, 2023, the OWID data was actually just piped from the Johns Hopkins University COVID-19 dashboard - there are several publications that can be cited about this work (Dong et al 2020 and Dong et al 2022). If after that date, the case data comes via the World Health Organization. Recognizing the original source is important as the two projects had different approaches to assigning dates to historical cases when historical revisions were made publicly available. These, in turn, impact when the daily cases appear in the case time series. Additionally, the historical case data on OWID is under revision so it is possible the associations reported in the manuscript may change if the analysis were repeated in the future. With a download date, individuals could look at commits to the OWID GitHub repository and use case data from the same date as in this manuscript.

Minor Comments

- The use of the oxford comma throughout the manuscript is inconsistent. Please screen and add or remove, depending on author preferences.

- Page 4, paragraph 1: There is an unmatched closed parenthesis (“)”) in this sentence.

- Page 4, paragraph 1: The statement “Third, virus fitness was associated with high infection numbers in all continents except Africa (no significance).” does not match the visualization in figure 2 where virus fitness has a negative association in North America.

- Page 5, paragraph 1: I believe “Supplementary Figure A1” is meant to refer to “Figure 1” and “Supplementary Figure A2” is meant to refer to “Figure 2”. Please confirm.

- Page 8, figure 3 caption: missing a parenthesis after “Simmonds et al”

- Page 9: Please use a “substantial” instead of “dominant” to refer to the ADAR signature. I think “dominant” is slightly confusing as the most dominant signature remains APOBEC.

- Page 10, figure 5: I want to note that I found this figure very interesting and well-presented. There are a few small items to fix. In panel B, would it be possible to change the country names away from all capitals and remove the underscores for North and South America? In panel D, the y axes dynamically fit the data for all continents except Asia. Can that be adjusted?

- Page 10: I don’t believe Ruis et al should be cited as “Ruis, Peacock et al”

- Page 11, section 2.5: The paragraph is interesting, but it doesn't need to be stated twice to be understood (it's currently duplicated).

- Page 17: Please be consistent with writing out “negative sense” or abbreviating as “- sense”

- Page 17: ROS has already been defined on page 7

- Page 17: I would recommend reading and citing PMID: 34061905 when discussing ROS. Ansari and Simmonds only found a ROS-like signature in coronaviruses and not other RNA viruses, which is interesting.

- Page 19: Be explicit as to whether the statement “Synonymous changes were much more likely to occur in ORF1a/b, which would be expected due to its size as the largest ORF,” is referring to counts of synonymous mutations or rates of synonymous to non-synonymous mutations. I could agree with the former, but don’t quite believe the latter.

- Page 19: Replace “bu” with “by”

- Page 19, paragraph 4: Would the authors be able to provide a bit more detail as to how sequences with “excessive mutations/deletions” were identified and screened out?

- Page 20, paragraph 4: There is a missing parenthesis at the end of “log(total mutations per week” (in the equation)

- Page 32, figure: Please change the y-axis to “Daily new cases”

Reviewer #2: Lamb et al employ the very large dataset of SARS-CoV-2 genomes to examine the factors driving case trajectories and the signatures underlying mutational patterns. They identify several variables that correlate with case numbers. The authors then extract three mutational signatures from SARS-CoV-2 mutational spectra, assign potential drivers and examine their dynamics through time. The potential impact of these signatures on VOC mutations is then examined. The authors finally examine the presence of the signatures in wastewater data. The three signatures the authors have extracted and their dynamics are novel and interesting. However, as outlined in more detail below, the assignment of these signatures to drivers is not sufficient, and the assignment of individual mutations to signatures based on their mutation type is not reliable. This leads to the authors making conclusions that are not supported by the data.

The three signatures that the authors have extracted are novel and interesting, as are the dynamics of these signatures through time. However, the assignment of these signatures to drivers based simply on their dominant substitution type is highly problematic. For example, there are a large number of processes that can cause C-to-T mutations (for example see PMID: 30982602). APOBECs are one such process but it is not sufficient to identify C-to-T mutations and jump straight to APOBECs as the cause. Likewise, there are other factors that can cause G-to-T mutations apart from ROS. Signature 2 consists of A-to-G and T-to-C mutations, which are symmetric so the authors assume ADAR; why would this signature not be consistent with a change in polymerase error profile or another mutagenic factor that causes both substitution types? The assignment of signatures to causes is a major issue with the manuscript currently as it drives a lot of the narrative, including the title. But there is not any support provided for these drivers over any other driver of these mutation types. It is reasonable to speculate that APOBECs, ADAR and ROS may contribute to the mutational burden of SARS-CoV-2 but there is not evidence to show that they are major drivers and the conclusions of the manuscript in this regard are therefore not supported. I’d suggest that the authors refer to the signatures throughout as, for example, ‘signature 1’ rather than ‘APOBEC-like’.

In Figures 4, 5 and 6, the authors assess the proportion of mutations associated with their extracted signatures. According to these plots and the author-assigned drivers of the signatures, we are led to believe that the viral polymerase has not made a single unrepaired replication error during the pandemic as all mutations are suggested to be driven by APOBEC, ADAR and ROS. While the proofreading exonuclease encoded by SARS-CoV-2 can recognise replication errors, it clearly won’t repair all mistakes and polymerase errors will contribute to mutational spectra. This again highlights why assigning signatures to APOBEC, ADAR and ROS without further support is a major issue.

There is a need for further validation of the patterns of mutational signatures through time. For example, the authors suggest that signature 2 causes the majority of mutations in Alpha in early 2021. If this were true, we’d expect to see a very high proportion of A-to-G and T-to-C mutations within this time period, as these are the dominant mutation types in signature 2. The authors include Figure A3 showing counts of unique substitutions per week split by mutation type (although this doesn’t seem to be referenced in the text). There doesn’t seem to be a large increase in the proportion of mutations that are A-to-G or T-to-C during early 2021, nor a decrease in C-to-T which would be expected if signature 1 is reducing within Alpha. While only a fraction of global sequences at this time are Alpha, I’d still expect to see some impact in this plot. To support the signature patterns, the authors should show similar plots to Figure A3 but broken down by variant and continent, to complement the signature plots in Figure 5c-d.

Section 2.5 examining the drivers of particular amino acid changes is highly problematic. Assigning mutations to signatures based on their mutation type is not possible. For example, C-to-T mutations are present in each of the three extracted signatures. The authors cannot therefore identify a C-to-T mutation and state that it was driven by signature 1, as there is a reasonable probability it could have been driven by signature 2 or 3, or by a currently unidentified signature. It’s also unclear why the authors would include G-to-A mutations in signature 1 here when G-to-A is a major contributor to signatures 2 and 3. The interpretations of this data are then highly problematic – for example, the authors state “Clearly, this APOBEC-like process was frequently active prior to the Alpha VOCs emer- gence”; this is based on the branch leading to Alpha having a proportion of C-to-T mutations. To assign these to signature 1 purely because they are C-to-T mutations is not possible. As outlined above, the drivers of these signatures are not supported, so this statement combines an unreliable assignment of mutations to signatures with an unreliable assignment of the driver of this signature and attempts to make a major conclusion about the processes contributing to variant generation. This is not reliable. Another example is “The high density of ADAR RBD mutations in a variant that has emerged as the ADAR signature has increased may suggest that the ADAR mutational process has driven the emergence of the Omicron variant.” The authors are making major conclusions here that are not supported by reliable data. These are examples but section 2.5 is largely formed of the same analysis and conclusions. It may be possible to generate a statistical model that can take a set of contextual mutations and provide some estimate of the processes that have contributed to their generation based on mutation types and their surrounding contexts but failing this, section 2.5 does not provide reliable insights.

The methods section is light on detail, but my reading of the methods is that the authors calculated mutational spectra by comparing individual sequences from GISAID against an “ancestral lineage reference sequence”. It is not clear from the methods whether this sequence is the ancestor of the same lineage the sequence is assigned to or the parental lineage as the authors use an example of sequences from lineage B.1 being compared against the reference for lineage B. Did the authors use one reference per lineage? Or were ancestral references only included for some lineages? Importantly, how were mutations counted? For example, if we take a cluster of related sequences within lineage B.1 that share 3 mutations that were acquired since the lineage B.1 ancestor, are these 3 shared mutations only counted once? Or are they counted in each of the sequences in the cluster? Each mutation should be counted on each phylogenetic branch on which the mutation occurred. Only counting each mutation once will undercount convergent mutations while counting each mutation in each sequence in which it occurs will result in highly inaccurate spectra. The methods here need to be clarified and, if necessary, updated.

How were the surrounding contexts of each mutation identified? Were these identified from the same “ancestral lineage reference” or from the Wuhan-Hu-1 sequence?

The authors don’t appear to have rescaled the calculated spectra by triplet availability in the genome, which is a necessary step to obtain reliable mutational spectra. For example, if the genome contains 1000 CCT contexts but only 100 GCG contexts, a C-to-T mutation would be far more likely to occur in a CCT context than a GCG context. Therefore to obtain a mutational spectrum that represents the actual mutational processes, it is necessary to divide the mutation counts by the number of occurrences of the starting triplet, as has been carried out in previous publications on SARS-CoV-2 spectra (for example PMID 37185044).

I’m not sure what the novel takeaways are from section 2.1 as all of the statements in this section (number of lineages, dominance of certain lineages, waves of different variants, infection rates, VOCs increasing infection rates) have been extensively covered in previous publications and public dashboards/resources.

Section 2.2 should include sensitivity analyses. There are significant challenges associated with the analyses in section 2.2 as testing and case reporting differ hugely by country. This may be possible to account for at the level of individual countries where there is good knowledge of the testing and reporting practices but at the continent level this appears to be a major challenge and it’s not clear that the data is reliable enough to enable effective modelling. Additionally, the authors only examine four predictor variables and exclude other factors that are very likely to have an effect, for example previous infection burden. These factors may contribute to the inability of the models to completely fit the data in the insets in Figure 1, for example the fit in Oceania appears poor. A sensitivity analysis to assess the impact of poor estimates of cases in some regions and the impact of missing variables would be needed to trust the output from this modelling (or citations to previous papers that have done this if such papers are available). Furthermore, the authors state that “fitted mean response values with predictor values as the input resembled the rise and fall of infection cases”. There is not currently a statistical assessment to support this and it is not clear that this is the case from examining the data in Figure 1; a statistical assessment of the model fits should be carried out to support this statement.

Minor comments

Results section 2.1, paragraph 1, line 5 – table A2 shows some Pango lineages and some WHO variant names, the “13 Pango lineages” should be updated to reflect this.

Figure 1 – from the legend, the bars show biweekly proportions of common lineages. Does the whitespace show the proportion of sequences from other lineages? Please specify.

Figure 1 – not all insets can be read, it would be useful to update the figure to make these clearer.

Methods section 4.1 – what were the cutoffs for “excessive mutations/deletions”? Please describe.

Methods section 4.2, paragraph 1, lines 2-3 – “We collected continuous dependent vari- ables that changed with every calendar date did not remain constant for finite amounts of time.” It’s not clear what this sentence means, is it possible to update?

Methods section 4.2, paragraph 1, lines 7-8 – missing vaccination data was handled as no vaccines being administered. Are these data points at times where it is reasonable to believe that no vaccines were administered (i.e. before vaccine rollouts in the respective countries)? This would be useful to state.

Methods section 4.2 – some more details on the estimation of virus fitness would be useful. How were the mutations in each lineage defined? How is the sum of fit mutations performed - is each “fit” mutation observed counted as 1 regardless of frequency in the population?

Methods section 4.7 – what was the NMF run on? Was this run on the set of spectra from all epidemic weeks? Please expand this section to make clear. A supplementary figure showing the reasoning for three signatures being chosen as optimum would be useful.

Figure 2 – while it’s clear which panel is which, panels A and B aren’t labelled in the figure currently.

Figure 3 – not all mutation types are represented in the signatures (for example A-to-C to T-to-G). These mutation types do occur within SARS-CoV-2 mutations (for example see PMIDs 37185044 and 37039557). This should be stated and included in the discussion around the activities of signatures through time, as it is likely that not all mutations are driven by the extracted signatures.

The use of mutation nomenclature is not always consistent – for example C->T or CT. Please update.

**Have the authors made all data and (if applicable) computational code underlying the findings in their manuscript fully available?**

Reviewer #1: **No: **Computational code should be made available on GitHub. If this has been done, it isn't referenced in the manuscript cover page.

Reviewer #2: **No: **The authors don't currently provide access to data. As the sequences are from GISAID, it will not be possible to include sequence data. However, the extracted signatures and code should be available and don't appear to be currently

PLOS authors have the option to publish the peer review history of their article (what does this mean?). If published, this will include your full peer review and any attached files.

Reviewer #1: **Yes: **Jeremy Ratcliff

Reviewer #2: No
---

## [Decision Letter · Decision Letter 1]

22 Nov 2023

Dear Prof. Robertson,

Thank you very much for submitting your manuscript "Is SARS-CoV-2’s evolutionary capacity mostly driven by host antiviral molecules?" for consideration at PLOS Computational Biology. As with all papers reviewed by the journal, your manuscript was reviewed by members of the editorial board and by several independent reviewers. The reviewers appreciated the attention to an important topic. Based on the reviews, we are likely to accept this manuscript for publication, providing that you modify the manuscript according to the review recommendations.

Sincerely,

Roger Dimitri Kouyos

Academic Editor

PLOS Computational Biology

Amber Smith

Section Editor

PLOS Computational Biology

Reviewer's Responses to Questions

**Comments to the Authors:**

Reviewer #1: I thank the authors for their thorough response to my previous major and minor comments. I find those specific concerns have been satisfactorily addressed. I have a few remaining minor comments in light of the changes to the manuscript:

- Please move the statement on lines 122-123 “Note, SARS-CoV-2 has an RNA genome… signature notations” to immediately after line 119 where thymine is first mentioned.

- I advocate for swapping the placement of figure 3 and supplementary figure 5 in the manuscript. The normalized values are a better representation of the mutational signature and do not detract from the overall takeaway of the impact of these substitutions on the three signatures. Doing so also strengthens the authors’ claims regarding the context-specific induction of the APOBEC-like signature. I recognize that my feelings regarding composition normalization are more stringent than that of the rest of the field and, for that reason, I treat this as a suggestion and not a requirement for the authors. Doing the replacement would require some editing of the discussion, where normalization results are generally presented after the raw values (which is sensible if the normalization is in the supplementary material; e.g., lines 363-364, 380-381, etc.)

- “Alphas” (sic) is missing a possessive apostrophe on line 420.

Reviewer #2: I’d like to thank the authors for their thorough responses to the previous round of comments. With these updates, I find the results convincing and exciting.

Discussion lines 384-386 – exposure of human DNA to ROS has shown quite strong context dependence, for example see figure 3B in PMID 30982602

Figure 5B – a legend showing the colours of the 3 signatures would be useful for clarity

Figure S7 – a legend showing the colour corresponding to each mutation type would be useful, as is currently included in Figure S8

**Have the authors made all data and (if applicable) computational code underlying the findings in their manuscript fully available?**

Reviewer #1: Yes

Reviewer #2: Yes

PLOS authors have the option to publish the peer review history of their article (what does this mean?). If published, this will include your full peer review and any attached files.

Reviewer #1: **Yes: **Jeremy Ratcliff

Reviewer #2: No

Figure Files:

Data Requirements:

Reproducibility:

References:

---

## [Editor Report · Decision Letter 2]

3 Jan 2024

Dear Prof. Robertson,

We are pleased to inform you that your manuscript 'Mutational signature dynamics indicate SARS-CoV-2's evolutionary capacity is driven by host antiviral molecules' has been provisionally accepted for publication in PLOS Computational Biology.

Best regards,

Roger Dimitri Kouyos

Academic Editor

PLOS Computational Biology

Amber Smith

Section Editor

PLOS Computational Biology

---

## [Editor Report · Acceptance letter]

19 Jan 2024

PCOMPBIOL-D-23-01052R2 

Mutational signature dynamics indicate SARS-CoV-2's evolutionary capacity is driven by host antiviral molecules

Dear Dr Robertson,

I am pleased to inform you that your manuscript has been formally accepted for publication in PLOS Computational Biology. Your manuscript is now with our production department and you will be notified of the publication date in due course.

With kind regards,

Zsofi Zombor
